# Melatonin Modulation of Radiation-Induced Molecular Changes in MCF-7 Human Breast Cancer Cells

**DOI:** 10.3390/biomedicines10051088

**Published:** 2022-05-07

**Authors:** Carolina Alonso-González, Cristina González-Abalde, Javier Menéndez-Menéndez, Alicia González-González, Virginia Álvarez-García, Alicia González-Cabeza, Carlos Martínez-Campa, Samuel Cos

**Affiliations:** 1Department of Physiology and Pharmacology, School of Medicine, University of Cantabria and Instituto de Investigación Sanitaria Valdecilla (IDIVAL), 39011 Santander, Spain; carolina.alonso@unican.es (C.A.-G.); cristina.gonzaleza@alumnos.unican.es (C.G.-A.); javier.menendezm@alumnos.unican.es (J.M.-M.); virginia.alvarez@unican.es (V.Á.-G.); coss@unican.es (S.C.); 2Unidad de Gestión Clínica Intercentros de Oncología Médica, Hospitales Universitarios Regional y Virgen de la Victoria and Instituto de Investigación Biomédica de Málaga (IBIMA)-CIMES-UMA, 29010 Málaga, Spain; agonzalez.bq@gmail.com

**Keywords:** melatonin, pineal gland, breast cancer, radiotherapy, radiosensitizer, radiotherapy adverse effects, anti-cancer agents, anti-cancer therapy sensitizer, anti-angiogenesis, improved individual outcomes, MCF-7 cells, gene expression, miRNA expression

## Abstract

Radiation therapy is an important component of cancer treatment scheduled for cancer patients, although it can cause numerous deleterious effects. The use of adjuvant molecules aims to limit the damage in normal surrounding tissues and enhance the effects of radiation therapy, either killing tumor cells or slowing down their growth. Melatonin, an indoleamine released by the pineal gland, behaves as a radiosensitizer in breast cancer, since it enhances the therapeutic effects of ionizing radiation and mitigates side effects on normal cells. However, the molecular mechanisms through which melatonin modulates the molecular changes triggered by radiotherapy remain mostly unknown. Here, we report that melatonin potentiated the anti-proliferative effect of radiation in MCF-7 cells. Treatment with ionizing radiation induced changes in the expression of many genes. Out of a total of 25 genes altered by radiation, melatonin potentiated changes in 13 of them, whereas the effect was reverted in another 10 cases. Among them, melatonin elevated the levels of PTEN and NME1, and decreased the levels of SNAI2, ERBB2, AKT, SERPINE1, SFN, PLAU, ATM and N3RC1. We also analyzed the expression of several microRNAs and found that melatonin enhanced the effect of radiation on the levels of miR-20a, miR-19a, miR-93, miR-20b and miR-29a. Rather surprisingly, radiation induced miR-17, miR-141 and miR-15a but melatonin treatment prior to radiation counteracted this stimulatory effect. Radiation alone enhanced the expression of the cancer suppressor miR-34a, and melatonin strongly stimulated this effect. Melatonin further enhanced the radiation-mediated inhibition of Akt. Finally, in an in vivo assay, melatonin restrained new vascularization in combination with ionizing radiation. Our results confirm that melatonin blocks many of the undesirable effects of ionizing radiation in MCF-7 cells and enhances changes that lead to optimized treatment results. This article highlights the effectiveness of melatonin as both a radiosensitizer and a radioprotector in breast cancer. Melatonin is an effective adjuvant molecule to radiotherapy, promoting anti-cancer therapeutic effects in cancer treatment. Melatonin modulates molecular pathways altered by radiation, and its use in clinic might lead to improved therapeutic outcomes by enhancing the sensitivity of cancerous cells to radiation and, in general, reversing their resistance toward currently applied therapeutic modalities.

## 1. Introduction

### 1.1. Radiotherapy in Cancer: Current Deficits in Breast Cancer Management

Applied to cancer patients, radiotherapy aims to reduce tumor volume by inducing cancer cell death or it can be alternatively used as a palliative treatment. It has been estimated that almost 80% of cancer patients receive treatments combining chemotherapy or hormonal therapy with radiation therapy [1,2]. Similar to most traditional cancer treatments, ionizing radiation triggers a variety of adverse side effects. For example, many intracellular signaling pathways are altered by the generation of reactive oxygen species, and the subsequent oxidative damage may lead to apoptosis or cell growth inhibition, as well as to stimulation of angiogenesis or epithelial-to-mesenchymal transition (EMT) transformation [3]. Moreover, radiation has deleterious effects on normal tissues. These reasons justify the importance of the discovery of new effective radiosensitizers that can protect normal tissues, enhance the curative of radiotherapy and help overcome resistance to radiation [4,5].

### 1.2. Multifunctional Role of Melatonin in Anti-Cancer Protection

Melatonin is a natural indoleamine mostly synthesized by the pineal gland in darkness, able to mitigate cancer at the initiation, progression and advanced stages [6]. Many research works conducted both in vivo and in vitro, as well as results from several clinical trials, have demonstrated that melatonin inhibits hormone-dependent mammary tumors by blocking estrogen signaling-mediated transcription, thus acting as a selective modulator of the estrogen receptor (SERM). Additionally, melatonin downregulates the enzymes necessary for estradiol synthesis, behaving as a selective estrogen enzyme modulator (SEEM) [7,8]. In recent years, a large number of research articles have concluded that the combination of melatonin with either chemotherapeutic agents or radiotherapy potentiates the effect of these treatments in many types of tumors [9].

### 1.3. Melatonin as an Effective Radiosensitizer in Cancer Cells and a Radioprotector of Non-Tumor Tissues

Specifically, when combined with radiotherapy, melatonin enhances cancer cell damage by increasing the radiation sensitivity of tumor cells, simultaneously protecting non-tumor tissues from the deleterious effects of this treatment. There are a variety of mechanisms by which melatonin potentiates the effects of radiation; among them, the pineal hormone enhances the acute cytotoxic effects of radiation by increasing the production of intracellular reactive oxygen species (ROS) [10]. In human breast cancer cells, the administration of melatonin as a previous step to radiation showed a radiosensitizing effect associated with a decrease in the synthesis of the enzymes responsible for estrogen biosynthesis [11]. Both in vivo and in vitro, the combination of melatonin and radiation diminished the efficiency of the DNA repairing mechanisms by downregulating proteins involved in double-strand DNA break repair [12,13]. The combination of melatonin and ionizing radiation resulted in a synergic inhibition of several pro-angiogenic factors [14].

### 1.4. Focus of the Current Study: Melatonin as a Helper in Anti-Cancer Radiation Therapy

The main goal of this work is based on previous studies demonstrating that melatonin administered with radiotherapy is able to enhance its therapeutic effects while protecting normal cells against the side effects of this treatment. Although many mechanisms have been proposed as involved in melatonin’s mediated radiosensitization, most of the molecular pathways by which melatonin enhances the global response of breast cancer cells to radiotherapy remain unrevealed. Therefore, our aim was to identify genes, miRNAs and signaling pathways altered by ionizing radiation in order to investigate which of these changes were enhanced and which were reversed by melatonin treatment. The starting hypothesis of our work was that melatonin behaves as a radiosensitizer that ameliorates the curative effects of radiotherapy by enhancing molecular changes induced by radiotherapy, leading to cancerous cell death, while protecting against undesirable changes triggered by radiation.

To accomplish this, we first determined the effect of ionizing radiation and melatonin on breast cancer cell proliferation. We next analyzed the influence of melatonin on the expression of multiple genes and miRNAs involved in breast cancer using a gene microarray (Human Breast Cancer RT^2^ Profiler^TM^ PCR Array) and a Human Breast Cancer miRNA microarray (MIHS-109ZA, Qiagen, Germantown, MD, USA). Genes and miRNAs whose expression was significantly modulated by melatonin were analyzed by specific RT-PCR studies. We also studied the effect of radiation and melatonin on intracellular signaling pathways. Finally, we evaluated the melatonin anti-angiogenic activity on the chick embryo chorioallantoic membrane model in vivo.

## 2. Materials and Methods

### 2.1. Cells and Culture Conditions

Cell culture experiments were performed using MCF-7 cells, a non-metastatic and triple-positive breast tumor cell line purchased from the American Tissue Culture Collection (MCF-7.atcc. HTC-22™) (Rockville, MD, USA). These cells were cultured as monolayers in 75 cm^2^ plastic flasks in Dulbecco´s Modified Eagle´s Medium (DMEM) (Lonza, Berna, Switzerland) supplemented with 10% fetal bovine serum (FBS), penicillin (20 units/mL) and streptomycin (20 µg/mL) (all from Lonza, Berna, Switzerland) and incubated at 37 °C with 5% CO_2_ to maintain a humid atmosphere. 

### 2.2. Ionizing Radiation

MCF-7 cells were exposed to X-ray irradiation using a YXLON SMART 200 tube (Yxlon International, Hamburg, Germany) at the Department of Radiology and Medical Physics of the University of Cantabria. Radiation was administered as an only dose in a 11.5 cm × 8.5 cm field size. The source-half-depth distance was initially calculated to obtain a constant dose rate of 0.92 Gy/min. The radiation dose used was 8 Gy, as previously described [13]. 

### 2.3. Cell Proliferation Assay

MCF-7 cells at 70–80% confluence, were initially cultured for 24 h in DMEM supplemented with 0.5% dextran-charcoal stripped FBS (csFBS) and incubated at 37 °C for 24 h to allow cellular attachment. Melatonin pre-treated cells were incubated for 7 days in DMEM supplemented with 10% FBS containing 1 nM melatonin prior to any radiation treatment. Cells were seeded at a density of 8 × 10^3^ cells per well and 24 h later, melatonin pre-treated and/or control MCF-7 culture plates were irradiated at a dose of 8 Gy. Control cells were removed from the incubator and placed for the same period of time into the irradiator, but without receiving any radiation. After 6 days, cell proliferation was measured by the MTT [3(4,5dimethylthiazol-2-yl)-2,5-diphenyl tetrazolium bromide] method, reading absorbance at 570 nm in a microplate reader (Labsystems Multiskan RC 351, Vienna, VA, USA). 

### 2.4. Total RNA Extraction and cDNA Synthesis

Total cellular RNA, including microRNA and other small RNA molecules, was isolated using the miRNeasy RNA kit (Qiagen, Germantown, MD, USA) following the manufacturer’s instructions. The quality and quantity of the RNA eluted were measured using a spectrophotometer (Nanodrop 1000 V 3.6, Thermo Fisher Scientific, Waltham, MA, USA). The ratio A_260_/A_280_ was used to assess RNA purity, considering samples with a ratio between 1.9–2.0 as pure RNA. For cDNA synthesis, the RT^2^ First Strand kit (Qiagen, Germantown, MD, USA) was employed, using 0.5 µg of total RNA as a template. First, samples were incubated in a thermocycler (MyCycler Gradient, Bio-Rad, Hercules, CA, USA) for 5 min at 42 °C to eliminate genomic DNA, followed by two incubations of 15 min at 42 °C and 95 °C for 5 min in a final volume of 20 µL. Afterward, 91 μL of RNA-free water was added to each reaction and the samples were kept on ice until proceeding with the real-time PCR (RT-PCR) protocol.

### 2.5. RT^2^ Profiler ^TM^ PCR Gene Expression Array

Pathway-focused gene expression profiling was performed using a 96-well human breast cancer PCR array (RT^2^ Profiler PCR array-PAHS-131ZA, Human Breast Cancer PCR Array, Qiagen, Germantown, MD, USA), which includes all the components to study the expression of 84 genes related to breast cancer pathways and different housekeeping and controls to allow data normalization. MCF-7 control or melatonin pre-treated cells (1 nM) were seeded into 6-well plates at a density of 8 × 10^5^ cells per well and incubated for 24 h prior to being irradiated (8 Gy). Six hours after irradiation, total cellular RNA was isolated and reverse transcribed, as described previously. The cDNA template was mixed with the appropriate amount of RT^2^ SYBR Green qPCR Mastermix (Qiagen GmbH, Hilden, Germany), aliquoted (25 µL) to each well of the same plate and then RT-PCR was performed in an MX3005P (Agilent, Santa Clara, CA, USA) following the manufacturer’s instructions. Cycling conditions were: 1 cycle at 95 °C for 10 min followed by 40 cycles using the following temperature profile, 95 °C for 30 s (denaturation) and 60 °C for 60 s (annealing/extension). Dissociation curves were performed to verify that only a single product was amplified. The Ct data for each gene were analyzed using the Qiagen RT^2^ Profiler PCR array data analysis software https://geneglobe.qiagen.com/us/analyze (12 January 2021). Data are represented as fold-changes between experimental groups and control cells.

### 2.6. Conversion of Mature microRNA into cDNA

In order to perform microRNA expression analysis, RNA samples were polyadenylated by poly(A) polymerase and subsequently converted into cDNA by a reverse transcription reaction using an oligo-dT primer with universal tag sequence on the 5′ end (miScript II RT Kit Qiagen, Germantown, MD, USA). For cDNA synthesis, 250 ng of total RNA were mixed with the different components according to the manufacturer’s instructions and were incubated in a thermocycler (MyCycler Gradient, Bio-Rad, Hercules, CA, USA) following a two-step reaction: 60 min at 37 °C and 5 min at 95 °C to inactivate reverse transcriptase activity. Finally, each sample was diluted in 200 µL of RNase-free water and stored at −20 °C until use.

### 2.7. Breast Cancer Pathway-Focused microRNA PCR Array

To study changes in microRNA expression through radiation and melatonin treatments, pathway-focused miRNA expression profiling was performed using a Human Breast Cancer microarray (MIHS-109ZA, Qiagen, Germantown, MD, USA). Specifically, this array consists of 84 mature miRNA forward primers arrayed in miRNome panels that appear to be somehow related to breast cancer. Each well contains all the components required to ensure that the quantitative PCR reaction generates a single gene-specific amplicon. The array also contains miScript PCR normalization controls, a reverse transcription control (miRTC) and a positive PCR control (PPC). The cDNA was mixed with the reagents provided by the miScript SYBR Green PCR Kit (Qiagen, Germantown, MD, USA), which contains a miScript universal primer as a reverse primer that allows the detection of mature miRNAs in combination with miScript-specific primers attached to the array plate wells (forward primer). For a 96-well plate, 25 µL of master mix were dispensed to each well and the plates were placed in a RT-PCR thermocycler (CFX96 Bio-Rad, Hercules, CA, USA) using the following protocol: an initial activation step of 15 min 95 °C with a ramp rate of 1 °C/s and a three-step cycling based on a denaturation step (15 s; 94 °C), an annealing step (30 s; 55 °C) and an extension step (30 s; 70 °C). The cycling step was repeated 40 times. Melting curves were plotted after the final cycling step to verify the unspecific amplification of primers. Data obtained by RT-PCR were analyzed using the Qiagen GeneGlobe online tool for microarray data management: https://geneglobe.qiagen.com/us/analyze (17 April 2021). Ct values were exported from the original RT-PCR software and transcribed into the GeneGlobe Analyze section, obtaining fold change data (based on ΔΔCt method), sample quality assessments and different plots. Data are represented as the fold-change relative to control cells. 

### 2.8. Analysis of Specific Gene Expression

The analysis of specific gene expression was carried out by RT-PCR following incubation of the cells for six hours after irradiation, as described above. Total cellular RNA was isolated and reverse transcribed as indicated above and RT-PCR was performed on an Mx3005P RT-PCR System (Agilent Technologies, Santa Clara, CA, USA) using the same temperature profile as indicated. Reactions were run in triplicate and melting curves were performed to verify that only a single product was amplified. Data are presented as the fold change between the experimental groups and the control cells. The primers used for amplification (Sigma Genosys Ltd., Cambridge, UK) are listed in Table 1. 

### 2.9. Analysis of Specific microRNA Expression

Selected microRNAs (miR-20a, miR-20b, miR-19a, miR-29a, miR-93, miR-17, miR-141, miR-15a, miR10a, miR10b and miR34a) (Qiagen, Germantown, MD, USA) were subjected to specific RT-PCR to further validate their expression profiles. RNA extraction samples were used to obtain cDNA via a reverse transcriptase reaction with the reagents provided by the miScript Primer Assays Kit (Qiagen, Germantown, MD, USA). RNase-free water was added to the mix to complete a total volume of 25 µL/per well. SNORD68 and SNORD95 microRNAs were used as normalization controls (Quiagen, Germantown, MD, USA). RT-PCR was performed in a thermocycler (CFX96 Bio-Rad, Hercules, CA, USA) using the following protocol: an initial activation step of 15 min 95 °C and a three-step cycling based on a denaturation step (15 s; 94 °C), an annealing step (30 s; 55 °C), and an extension step (30 s; 70 °C), repeating the cycling step 40 times. Melting curves were plotted after the extension step of each cycle to verify the unspecific amplification of primers.

### 2.10. Phosphokinase Screening

To determine the sensitizing effects of melatonin on radiation-induced changes in multiple kinases and their protein substrates, we used a membrane-based sandwich immunoassay Human Phospho-Kinase Array Proteome Profiler^TM^ (R&D Systems, Minneapolis, MN, USA). This array allows the simultaneous detection of the relative phosphorylation levels of 37 kinases. MCF-7 cells either pre-treated or not with melatonin (1 nM) were seeded into 6-well plates at a cellular density of 8 × 10^5^ cells and 24 h later were irradiated (8 Gy). After a 4 h incubation period, cells were washed twice with chilled PBS, lysed and mixed for 30 min in a rocking platform at 4 °C. Cell lysates (containing 500 µg of protein) were then added to the previously blocked nitrocellulose membranes and incubated overnight at 2–8 °C on a rocking platform. After performing two PBS washes, antibody detection cocktails were added to the membranes and incubated for 2 h at room temperature. Once the incubation step was finished, streptavidin-HRP-conjugated secondary antibodies were used for chemiluminescent detection, generating a signal directly proportional to the amount of protein bound to the membrane. Finally, the membranes were exposed to a film using a chemical reagent mixed with hydrogen peroxide and luminol. Pixel densities on the X-ray film were collected and analyzed using a LI-COR Odyssey IR Imaging System V3.0 (LI-COR Odyssey Biosciences, Lincoln, NE, USA). 

### 2.11. Western Blot Analysis

MCF-7 control and melatonin pre-treated cells were seeded and irradiated, as described previously. Four hours after radiation, confluent cells were washed twice with PBS and lysed with RIPA buffer containing 1% protease inhibitor cocktail. The protein concentration was assessed by the Bradford colorimetric method with a standard curve generated using BSA (Bovine Serum Albumin, Sigma-Aldrich, Burlington, MA, USA). The cellular proteins (25 µg) were separated by SDS-PAGE, using 3 µL of NZYBlue (NZYtech, Lisbon, Portugal) molecular weight marker as a reference, transferred to a polyvinylidene fluoride (PVDF) membrane (Bio-Rad, Hercules, CA, USA) and subjected to a constant voltage of 100 V at 4 °C for 100 min. Protein transference efficacy was assessed by membrane staining with Ponceau S red and gel staining with Coomassie Blue. For membrane blockage, a solution of 3% BSA in TBS-T (Tris-HCl 10 mM, pH 7.6, NaCl 150 mM, Tween 20 0.05%) was added to the membranes for 1 h at room temperature. Afterwards, the membranes were incubated with the selected primary antibody diluted in blocking solution at 4 °C overnight under rocking and finally with a fluorescent secondary antibody for 1 h at room temperature. The blots were stripped and re-probed with actin antibody (Sigma, St. Louis, MO, USA) to evaluate loading. The following antibodies were used: rabbit anti-AKT, rabbit anti-P-AKT (S473), rabbit anti-P70S6K, rabbit-anti-P-P70S6K (Thr389) and mouse anti-actin, all purchased from Cell Signalling (Danvers, MA, USA). Protein bands were detected by incubation with Anti-rabbit IRDye-680RD (LI-COR Odyssey Biosciences, Lincoln, NE, USA) or Anti-mouse IRDye-800CW (LI-COR Odyssey Biosciences, Lincoln, NE, USA). Fluorescence signals were detected using LI-COR Odyssey IR Imaging System V3.0 (LI-COR Odyssey Biosciences, Lincoln, NE, USA). To study differences in optical density bands, we used Image Studio (LI-COR Odyssey Biosciences, Lincoln, NE, USA) and ImageJ software. 

### 2.12. Chick Chorioallantoic Membrane (CAM) Model of Angiogenesis

As an in vivo model for studying angiogenesis, we used the CAM assay as previously described [15]. Fertilized eggs were incubated at 37 °C in a humidified incubator for 3 days. Hypodermic needles were used to remove 4 mL of egg albumin to allow detachment of the developing CAM shell. At day 4, the shells were covered with transparent adhesive tape and a small window sawed with scissors on the broad side directly over the avascular portion of the embryonic membrane. At day 11, MCF-7 control or melatonin pre-treated cells were irradiated at 8 Gy and embedded in alginate beads containing 1 × 10^6^ cells dissolved in PBS to be grafted on the CAM. After 72 h, new blood vessels converging toward the alginate were counted at 5× magnification under a STEMI SR stereomicroscope equipped with a 100 mm objective with adapter ring 47,070 (Zeiss) and fixed with 7% buffered formalin and photographed.

### 2.13. Statistical Analysis

Results from cell proliferation, gene expression and miRNAs are expressed as mean ± standard error of mean (S.E.M.) from three independent experiments. Two different protein extracts were analyzed and data were represented as a percentage of control (non-treated cells). Statistical differences between groups were processed by one-way analysis of variance (ANOVA) followed by the Student–Newman–Keuls test, with *p* ≤ 0.05 was considered to be statistically significant. Image Studio 5.2 and ImageJ software were used for protein analysis, CFX Maestro 2.0 and Qiagen GeneGlobe Analyze tool for RT-PCR data analysis and GraphPad Prism5 software for plotting data and statistics.

## 3. Results

### 3.1. Effects of Melatonin and Ionizing Radiation on the Proliferation of MCF-7 Cells

As a first approach, we aimed to confirm that melatonin enhances the anti-proliferative effects of ionizing radiation in the hormone-dependent MCF-7 cell line. Previous results from our laboratory established 8 Gy as the optimal radiation dose for MCF-7 cells. As shown in Figure 1, the inhibition of cell proliferation with radiation alone was 33% after 6 days of incubation. As expected, the greatest inhibition of cell proliferation was found when cells were pre-treated with melatonin (1 nM) before being radiated, showing a reduction of 62% after 6 days when compared to non-radiated cells.

### 3.2. Effects of Ionizing Radiation and Melatonin on the Expression of Cancer-Related Genes

Since the changes in gene expression produced on radiated MCF-7 cells and their modulation by melatonin are quite unknown, we used the human breast cancer RT^2^ Profiler PCR Array to assess the expression changes in MCF-7 cells upon treatment with ionizing radiation (8 Gy) either alone or after pre-treatment with a physiological dose of melatonin (1 nM). This array allows the simultaneous analysis of 84 genes, as described in the materials and methods. As summarized in Table 2, radiation stimulated the expression of 22 genes and downregulated 47 genes. When cells were pre-treated with melatonin, 23 genes were upregulated, and more than 60 were downregulated.

We next selected 25 genes for further validation by specific real-time RT-PCR analysis, establishing as selection criteria an expression level change of at least 1.5-fold either with ionizing radiation alone or in combination with melatonin compared to vehicle-treated (control) cells and/or, its well-established relevance in the genesis and development of breast cancer.

In the first group of analyzed genes, we found that ionizing radiation significantly inhibited the expression levels of MUC-1, c-MYC, BIRC5, BCL-2 and ABCB1. Pre-treatment with melatonin further enhanced the inhibitory effect of radiation (Figure 2a). Conversely, melatonin reverted the inhibitory effect of radiation on PTEN and NME1 expression (Figure 2b).

Another subset of genes stimulated by ionizing radiation and melatonin pre-treatment potentiated this effect: RASFF1, CST6, TP53, CDKN1A, BAX2, BAD, RB1 and PGR (Figure 3a). We also found eight genes whose expression was induced by ionizing radiation, whereas the pre-treatment with melatonin counteracted the stimulatory effect: SERPINE, PLAU, SNAI2, SFN, ERBB2, AKT, N3RC1 and ATM (Figure 3b). The expression of APC was not affected by radiation and inhibited by melatonin. Finally, GATA3 was inhibited by radiation and melatonin did not have any significant effect (Figure 3c).

### 3.3. Effects of Ionizing Radiation and Melatonin on the Expression of Cancer-Related miRNAs 

To study the effects of melatonin and radiation on miRNA expression in MCF-7 cells, we employed the Human Breast Cancer microarray (MIHS-109ZA, Qiagen, Germantown, MD, USA). Establishing as selection criterion a fold-change of ± 2, melatonin pre-treatment showed a downregulation of 27 miRNAs and an upregulation of 6 miRNAs when compared to control cells (Figure 4a). In a similar way, radiation alone modified the expression of 33 miRNAs (29 downregulated and 4 upregulated), as shown in the heatmap (Figure 4b). Finally, in those cells that were pre-treated with melatonin for one week before being irradiated, 37 miRNAs were downregulated and 6 upregulated compared to non-radiated control cells (Figure 4c). Due to their relevance in breast cancer onset and progression, a set of 11 miRNAs (miR-20a, miR-20b, miR-17, miR-141, miR-15a, miR-19a, miR-29a, miR-93, miR-10a, miR-10b and miR34a) (Figure 4d), were selected to be studied specifically. 

MCF-7 cells exposed to ionizing radiation showed decreased expression of miR-20a, miR-20b, miR-19a, miR-29a, miR-93, miR-10b and miR-10a. When cells were pre-treated with melatonin, an even more pronounced inhibition of the expression of miR-20a, miR-20b, miR-19a, miR-29a, miR-93, miR-10a and miR-10b was obtained, whereas the indoleamine alone significantly changed the expression of miR-10a (Figure 5a). Quite surprisingly, ionizing radiation stimulated the expression of miR-17, miR-141 and miR-15a (Figure 5b). Importantly, melatonin treatment prior to radiation counteracted this stimulatory effect. Finally, radiation alone enhanced the expression of miR-34a, and pre-treatment with melatonin strongly stimulated this effect (Figure 5c).

### 3.4. Effects of Ionizing Radiation and Melatonin on Kinase Intracellular Regulators

The next step was to investigate the effect of ionizing radiation and melatonin on the activation/inactivation status of protein kinases by phosphorylation. These sets of experiments were accomplished by using the Proteome Profiler^TM^ Array (R&D Systems, Minneapolis, MN, USA) to determine the phosphorylation status of multiple protein kinases involved in multiple intracellular signaling pathways. As shown in Figure 6a, the most relevant results found in the array panels corresponded to two proteins belonging to the mTOR signaling pathway: Akt (Ser473), which directly phosphorylates mTOR, and the ribosomal protein p70S6 kinase (Thr389), one of the downstream targets of mTORC1, leading to cell survival and proliferation when activated. Radiation alone induced a reduction in the relative phosphorylation levels of both proteins compared to control cells. These inhibitory effects were also potentiated when cells were pre-treated with melatonin prior to radiation (Figure 6b). 

To validate this data, we performed protein quantification studies by western blot. Rather surprisingly, radiation induced a significant increase in total Akt levels compared to the control cells. Melatonin pre-treatment counteracted this stimulatory radiation effect (Figure 6c). With respect to the Akt phosphorylated version (p-Akt), neither radiation nor melatonin modified its levels. Nevertheless, melatonin-pre-treatment prior to radiation showed a significant decrease in Akt activation compared to control cells (Figure 6d). Regarding p70S6K, radiation slightly increased its total protein levels, and this effect was counteracted by melatonin pre-treatment (Figure 6e). However, none of the treatments significantly changed p70S6K phosphorylation (Figure 6f).

### 3.5. Influence of Melatonin and Radiation on Newly Formed Blood Vessels in an In Vivo Angiogenesis Assay 

The newly formed blood vessel branch points were studied in the chick chorioallantoic membrane assay (CAM), an in vivo model of angiogenesis. As shown in Figure 7a and quantified Figure 7b, when MCF-7 cells were exposed to ionizing radiation or pre-treated with melatonin prior to implantation, a potent anti-angiogenic response was observed. In addition, the combined use of both treatments on the cells prior to its implantation in the eggs caused the formation of the vessels to be even lower. 

## 4. Discussion

### 4.1. Current Deficits in Radiation Therapy

Radiation therapy is a widely used treatment with curative or palliative purposes, using ionizing radiation; more than 50% of people diagnosed with cancer will receive radiotherapy [16,17,18]. The main purpose of this treatment is to kill malignant cells, but unfortunately, ionizing radiation can also induce numerous deleterious effects, both in the surrounding normal tissues and in survival cancer cells, by enhancing their resistance or stimulating processes such as angiogenesis or epithelial-to-mesenchymal transition that can lead to an even more malignant tumor phenotype [3,19]. 

### 4.2. Usefulness of Radiosensitizers 

For this reason, it is essential to develop new protocols based on the use of radiosensitizers that can enhance the effect of radiation on tumor cells without increasing the side effects in healthy tissues. Moreover, radiosensitizers should contribute to overcoming resistance to radiation [2,20]. Although many of the molecular modifications that breast cancer cells undergo after ionizing radiation have been described, many intracellular modifications remain to be characterized. 

### 4.3. Melatonin as a Radiosensitizer in Breast Cancer

Melatonin behaves as a radiosensitizer since its administration with radiotherapy enhances the therapeutic effects and protects normal cells against the side effects of this treatment. There are a variety of mechanisms that explain how melatonin potentiates the effects of radiation. Melatonin stimulates intracellular ROS, whose accumulation plays an important role in the mitochondrial apoptosis pathway [21]. Melatonin is also known to sensitize human breast cancer cells to the effects of ionizing radiation through an increase of p53 leading to induction of apoptosis [11,22]. Melatonin administration prior to ionizing radiation increases the radiosensitizing effects through regulation of enzymes involved in estradiol biosynthesis; thus, radiation decreases the levels of aromatase, sulfatase and 17β-Hydroxysteroid dehydrogenase 1, and melatonin pre-treatment further enhanced this inhibitory effect triggered by radiation [11]. Similar results were obtained in co-cultures of human endothelial cells and breast cancer cells [15] and in breast adipose fibroblasts [14]. Treatment with melatonin prior to radiation decreases the effectiveness of DNA repair by decreasing the RAD51 and DNA-protein kinase levels [13]. Similar effects were observed in vivo in xenograft tumor models of colorectal cancer in mice exposed to γ-ray radiation [12]. Finally, melatonin potentiates the antiangiogenic actions induced by radiation while neutralizing the angiogenic actions [15]. With the aforementioned precedents, the aim of this work was to deepen the characterization of the molecular mechanisms triggered by ionizing radiation in breast cancer cells and its modulation by melatonin.

### 4.4. Melatonin Enhances the Anti-Proliferative Effects of Ionizing Radiation in MCF-7 Cells

First, we tested the effect of ionizing radiation and melatonin on the proliferation of MCF-7 cells. As previously reported [11], cells pre-treated with melatonin at physiological concentrations (1 nM) prior to radiation showed a significant reduction in cell proliferation.

### 4.5. The Protective Role of Melatonin Based on the Regulation of Gene Expression Altered by Ionizing Radiation

In the next step, we examined the gene expression profile of breast cancer-related genes in MCF-7 cells treated with ionizing radiation alone or in combination with melatonin pre-treatment. We further analyzed the expression of 25 genes using specific RT-PCR. In some cases, melatonin and ionizing radiation cooperated to inhibit gene expression; that was the case of MUC1, an oncoprotein often overexpressed in cancer and that represses activation of the p53 gene [23]. It has been recently described that the administration of anti-MUC1 C antibody-conjugated, gadolinium-based nanoparticles improved the efficacy of radiation therapy [24]. c-MYC, a downstream target of ERK5, is involved in EMT, cancer progression and poor patient survival [25]. BIRC5 (survivin), an anti-apoptotic factor involved in multidrug resistance in breast cancer [26]. An increase in radiation resistance associated with survivin treatment has been reported after exposure to ionizing radiation [27]. BCL-2 expression was also inhibited by the combination of melatonin and radiotherapy, thus enhancing the sensitivity of colon cancer cells to ionizing radiation [28] and increasing cell death in irradiated sarcoma cells [29]. The last gene whose expression was repressed by radiation and further inhibited by melatonin was ABCB1, a protein involved in multidrug resistance, whose expression was induced by high doses of radiation but subsequently decreased significantly by low doses in colorectal cancer cells [30].

In two particular cases (PTEN and NME1), ionizing radiation had an unexpected and unwanted inhibitory effect, and melatonin was able to revert the inhibition of their expression. PTEN is a tumor suppressor gene whose silencing in breast cancer has been related to chemoresistance [31]. Moreover, the downregulation of PTEN expression has been related to radioresistance in non-small cell lung cancer cells [32]. On the other hand, ionizing radiation inhibited the expression of NME1, a metastasis suppressor in breast cancer cells [33], and this inhibitory effect was counteracted by melatonin.

We found that the expression of eight genes was stimulated by ionizing radiation and melatonin further enhanced this effect: TP53 (p53) and CDKN1A (p21), two classical tumor suppressor genes with high levels that correlate to better response to therapy [34]. We had already described that melatonin plus radiation induced a 2-fold change in p53 expression as compared to cells only radiated [11]. It has been proposed that the status of p53 expression in cancer cells has a direct relationship with radiotherapeutic efficacy [35]. With respect to p21, it has been described that it is up-regulated by irradiation, leading to apoptosis and radio-sensitization [36]. BAX-2 and BAD, two pro-apoptotic members were also up-regulated by ionizing radiation and this stimulatory effect was enhanced when melatonin was administered prior to radiation. BAD heterodimerizes with anti-apoptotic proteins, such as Bcl-2, inactivating them, thus allowing BAX2-dependent apoptosis [37]. An increase in Bax levels has been described in locally advanced prostate cancer patients in response to intraoperative radiotherapy [38]. RB1 (retinoblastoma) is a tumor suppressor gene whose loss drives tumorigenesis in limited types of cancer, but importantly, its function is often suppressed during tumor progression [39]. Interestingly, CDK4/6 inhibition increased the radiosensitivity of both estrogen-positive and triple-negative breast cancers in RB wild-type but not in RB-null cells [40]. PGR (progesterone receptor) is usually overexpressed in breast carcinomas. It has been described that the acquisition of radioresistance in MCF-7 cells resulted in a loss of PgR levels [41]. In our hands, melatonin enhanced the stimulatory effect of ionizing radiation on PGR expression. An identical effect was observed in the expression of CST6 (cystatin), a cysteine proteinase inhibitor whose loss is often associated with breast-to-bone metastasis [42]. The last gene whose expression was increased by radiation and melatonin was RASSF1, a tumor suppressor gene that is frequently inactivated in cancer by methylation of its promoter [43]. In oral squamous cell carcinoma, radioresistance was associated with RASSF1 promoter methylation and loss of RASSF1 expression was found to be significantly associated with poor disease-free survival [44].

Melatonin significantly counteracted the stimulatory effect of ionizing radiation on the expression of the following genes: SERPINE, also known as plasminogen activator inhibitor 1 (PAI-1), an oncogene that enhances the radioresistance and aggressiveness of non-small cancer cells (NSCLC). Moreover, PAI-1 secreted from radioresistant NSCLC cells reduced the radiosensitivity of nearby cells in a paracrine manner [45]. PLAU (urokinase plasminogen activator), involved in invasion and metastasis, showed augmented levels in cervical cancer cell lines [46]. SNAI2 (Slug), a transcription factor whose expression is involved in epithelial-to-mesenchymal transition. Slug levels seem to be increased in triple-negative breast cancer radioresistant cells [47]. ERBB2 (HER2/NEU) is frequently highly expressed in breast cancers and participates in cell signaling pathways leading to proliferation, angiogenesis and metastasis [48]. Clinical doses of ionizing radiation are known to stimulate the invasive properties of ErbB2-positive breast cancer cells and, moreover, radiation resulted in activation of ErbB2, which, through activation of FoxM1, promoted invasion of breast cancer cells [49]. SFN (stratifin 14-3-3) was first described as a tumor suppressor gene silenced in most breast tumors, but it has also been described as determinant for breast tumor invasion and associated with poor clinical outcome [50]. AKT is a critical ser/thr kinase that regulates multiple cellular functions, some of them important for cancer progression, such as proliferation, survival or apoptosis. Activation of AKT by phosphorylation correlates with poor prognosis and has been previously described in MCF-7 cells [51,52]. Overexpression of activating transcription factor 3 (ATF3) increased radioresistance by inducing AKT phosphorylation in breast cancer cells [53]. N3RC1 encodes for the glucocorticoid receptor, which, when activated, inhibits p53-dependent apoptosis in human mammary epithelial cells overexpressing the MYC oncogene [54]. Lastly, ATM (ataxia telangiectasia mutated) has been shown to be activated after ionizing radiation and, as a consequence, the cellular prion protein (PrPC) expression was increased, leading to radioresistance [55].

APC (adenomatous polyposis coli), initially described as a tumor suppressor gene, has been shown to co-immunoprecipitate with MUC1 in human breast tumors and metastasis [56]. In our hands, ionizing radiation did not alter the levels of APC and melatonin inhibited its expression. GATA3, a factor acting as a MUC1 transcriptional regulator in breast cancer cells [57], was downregulated by ionizing radiation and the addition of melatonin did not modify its transcription. 

Altered gene expression of genes involved in angiogenesis, EMT and metastasis often leads to the spread of the tumor and acquired chemoresistance and radioresistance. In estrogen-dependent breast cancer cells, radiation kills cancer cells principally by promoting DNA damage; however, at the same time, radiation also activates pro-survival molecular signaling pathways [58]. In the case of chemotherapy, treatment with agents such as doxorubicin leads to enhanced activation of pro-survival factors and increased invasion and migration as measured in three-dimensional spheroid invasion assays [51]. Interestingly, resistance mechanisms reducing doxorubicin sensitivity in MCF-7 cells are dependent on extracellular matrix proteins and sensitivity to doxorubicin is altered when cells are cultured in a 3D architecture, mimicking a tumor hypoxic state [59,60].

### 4.6. The Protective Role of Melatonin Is Also Based on the Regulation of miRNA Expression Altered by Ionizing Radiation

Next, we studied the effects of melatonin and radiation on the miRNA expression profile in MCF-7 cells. When cells were pre-treated for one week with melatonin or exposed to ionizing radiation, 33 out of the 84 total miRNAs were modified. Interestingly, changes in 44 miRNAs were observed when both treatments were combined. Due to their relevance in breast cancer pathogenesis, we selected a specific set of 11 miRNAs to validate these data: miR-20b, miR-93, miR-17, miR-20a, miR-29a, miR-15a, miR-19a, miR-141, miR-10a, miR-10b and miR-34a. Most of these miRNAs belong to highly conserved microRNA clusters, with increased expression in a wide spectrum of cancers, including breast cancer. The miR-17/92 cluster, involved in angiogenesis and considered to have oncogenic properties in breast cancer, encodes for six mature miRNAs, comprising miR-17, miR-20a and miR-19a as the most oncogenic ones [61]. In addition, there are two paralogue gene clusters, miR-106a-363 and miR-106b-25, encoding for miR-20b, miR-93 and miR-19a [62]. miR-10a and miR-10b belong to the miR-10 microRNA precursor family [63].

Radiation inhibited the expression of 7 miRNAs and the inhibition was even higher when melatonin was administered prior to radiation: miR-20a, miR-19a, miR-93, miR-20b, miR-29a, miR10a and miR10b.

miR-20a promotes proliferation and metastasis in many types of cancer and has angiogenic effects in breast cancer [64]. Another miRNA downregulated by ionizing radiation and melatonin was miR-19a, significantly upregulated and secreted from bone-tropic estrogen receptor (+) breast cancer cells, as a mechanism involved in osteolytic bone metastasis [65]. miR-93 is a pro-angiogenic miRNA contributing to the epithelial-to-mesenchymal transition in triple negative breast cancer [66]. miR-29a is upregulated in breast cancer stem cells and its overexpression promotes migration and invasion [67]. miR-20b, a member of the miR-106a-363 cluster, is upregulated in the brain metastasis of breast cancer patients in comparison to primary tumors [68]. Importantly, miR-20b is upregulated in radioresistant esophageal carcinoma cells, which indicates that it plays a role in tumorigenesis [69].

Rather surprisingly, three miRNAs were upregulated in response to ionizing radiation and importantly, melatonin reverted this effect: miR-17, miR-141 and miR-15a. miR-17 expression correlates with breast cancer stage, estrogen and progesterone receptor and lymph node status, and its upregulation promotes breast cancer migration and invasion [70]. High expression levels of miR-141 have been found in aggressive breast carcinomas and have been associated with shorter overall survival [71]. In vivo, knockdown of miR-141 inhibited the formation of novel breast cancer metastasis in the brain, whereas ectopic expression of miR-141 in the MDA-MB-231 cell line stimulated brain metastatic colonization [72]. There is some controversy about the role of miR-15a. It has been reported that miR-15a enhances radiation sensitivity in lung cancer by regulating the TLR1/NF-κB signaling pathway [73], but also that its inhibition in response to high dose radiation (10 Gy and above) decreased cell proliferation, triggered apoptosis and inhibited angiogenesis in a murine colorectal carcinoma model [74]. MicroRNA-34a is up-regulated in MCF-7 but not in T47D cells (lacking p53) in response to irradiation [75]. This microRNA behaves as a tumor suppressor, inhibiting epithelial-to-mesenchymal transition, migration and invasion [76]. According to our results, radiation stimulated the expression of this miRNA, and melatonin enhanced this effect in a remarkable way.

### 4.7. Melatonin Regulates Kinase Intracellular Regulators in Radiated MCF-7 Cells

Next, we focused our attention on the protein kinases that might be altered by radiation and melatonin. In a proteome profiler assay, we found that melatonin treatment before radiation induced an inhibition of Akt and p70S6 kinase phosphorylation compared to radiated cells, suggesting a melatonin protective effect, as aberrant activation of the PI3K/Akt/mTOR/p70S6K pathway, leads to cell growth, proliferation, survival and migration [77].

### 4.8. Melatonin and Radiation Cooperate in the Inhibition of the Formation of New Blood Vessels In Vivo

Additionally, we studied ionizing radiation and melatonin antiangiogenic activity in the Chorioallantoic Membrane Assay (CAM assay) as a model for angiogenesis. Previously, we demonstrated that melatonin has an inhibitory effect on angiogenesis when administered with chemotherapy [78]. Here, we show that both radiation and melatonin diminished the vascular area in this in vivo model, but when cells were treated with melatonin prior to radiation, the inhibitory effect was significantly higher.

## 5. Conclusions

In summary, the aim of this work was to contribute to deciphering the molecular changes induced by ionizing radiation in breast cancer cells, to investigate the role of melatonin potentiating some of them and counteracting some others. Indeed, we found that melatonin enhanced the effect of radiation in gene expression, Akt inactivation, miRNA expression, and new vascularization in vivo in many cases, but we also verified that radiation triggers undesirable deregulation in several cases, and importantly, melatonin was able to exert a protective role against them. These results confirm the potential of melatonin as an effective adjuvant to radiotherapy in breast cancer treatment and, as a whole, have allowed us to deepen our knowledge of the molecular changes that take place in tumor cells in response to ionizing radiation, to identify some of these potentially dangerous changes that can lead to proliferation, invasion, transition epithelial–mesenchymal, survival, metastasis and resistance, and they reaffirm the need to assay melatonin in clinical trials, seeking to corroborate its usefulness as an adjuvant to be used together with radiotherapy and chemotherapy. At this moment, there are very few clinical trials investigating the therapeutic usefulness of associating melatonin and radiotherapy in breast cancer patients. We are completely convinced that the use of melatonin as an adjuvant in cancer treatments should be implemented to improve individual results and the profitability of clinical services provided to cancer patients.

## Figures and Tables

**Figure 1 biomedicines-10-01088-f001:**
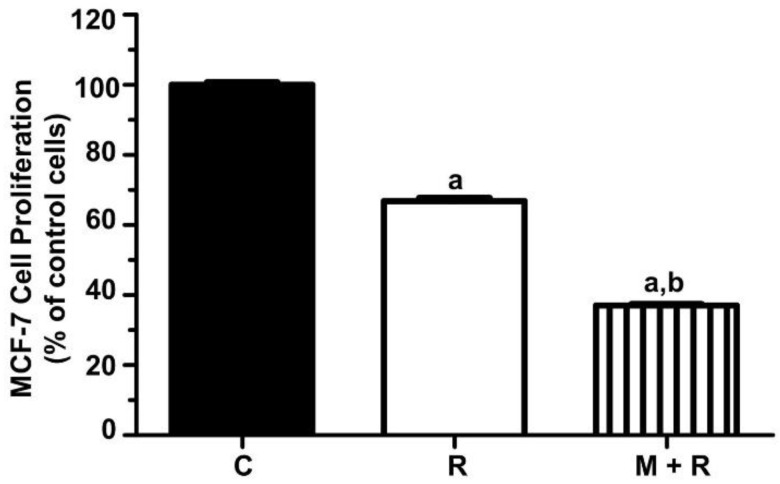
Effect of melatonin pre-treatment and ionizing radiation on MCF-7 cell proliferation. MCF-7 cell proliferation was measured by the MTT method six days after irradiation. Data are expressed as a percentage of control non-radiated cells (mean ± S.E.M.). a, *p* ≤ 0.001 vs. C; b, *p* ≤ 0.001 vs. R. C: Control; R: Radiated cells (8 Gy); M + R: Melatonin pre-treated and radiated cells.

**Figure 2 biomedicines-10-01088-f002:**
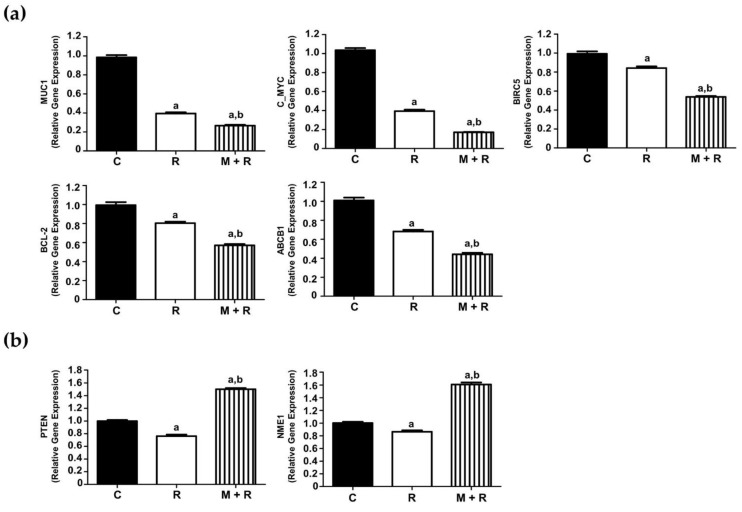
Effects of melatonin and radiation treatments on breast cancer-related gene expression in MCF-7 cells. (**a**) RT-PCR analysis of selected genes (MUC-1, c-MYC, BIRC5, BCL-2 and ABCB1) where the pre-treatment with melatonin (1 nM) further enhanced the inhibitory effect of radiation (8 Gy); (**b**) RT-PCR analysis of selected genes (PTEN, NME1), where melatonin pre-treatment reverted the inhibitory effect of radiation. All data are expressed as fold-changes relative to non-radiated (control) cells (mean ± S.E.M.). a, *p* ≤ 0.001 vs. C; b, *p* ≤ 0.001 vs. R. C: Control; R: Radiated cells (8 Gy); M + R: Melatonin pre-treated and radiated cells.

**Figure 3 biomedicines-10-01088-f003:**
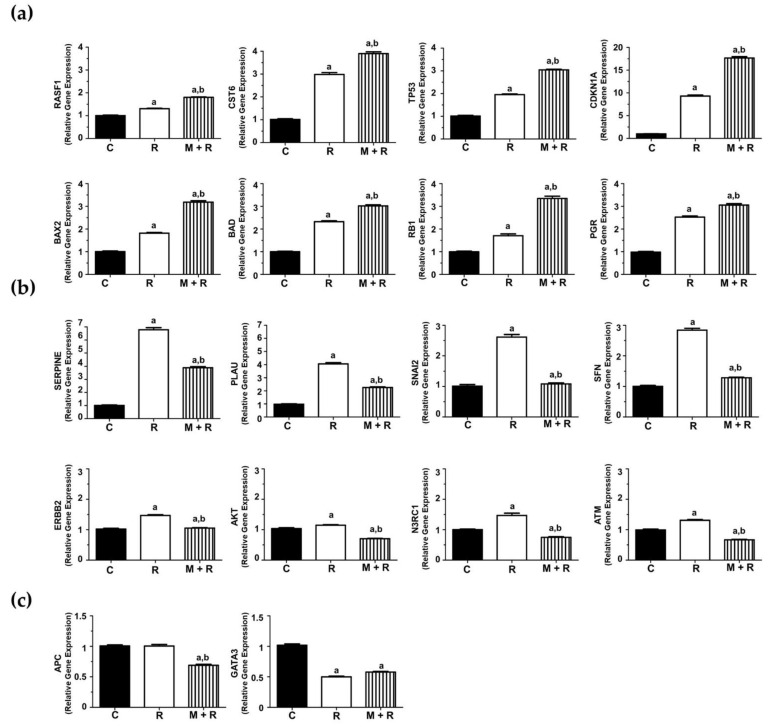
Effects of melatonin and radiation treatments on breast cancer-related gene expression in MCF-7 cells. (**a**) RT-PCR analysis of selected genes (TP53, CDKN1A, RB1, BAX2, BAD, RASFF1, CST6, PGR and CST6) where the pre-treatment with melatonin (1 nM) potentiated the stimulatory effect of radiation (8 Gy); (**b**) RT-PCR analysis of selected genes (SERPINE, SNAI2, SFN, ERBB2, AKT, N3RC1, PLAU and ATM) whose expression was induced by radiation whereas melatonin pre-treatment counteracted this stimulatory effect; (**c**) RT-PCR analysis of APC and GATA3 gene expression. All data are expressed as fold-changes relative to non-radiated (control) cells (mean ± S.E.M.). a, *p* ≤ 0.001 vs. C; b, *p* ≤ 0.001 vs. R. C: Control; R: Radiated cells (8 Gy); M + R: Melatonin pre-treated and radiated cells.

**Figure 4 biomedicines-10-01088-f004:**
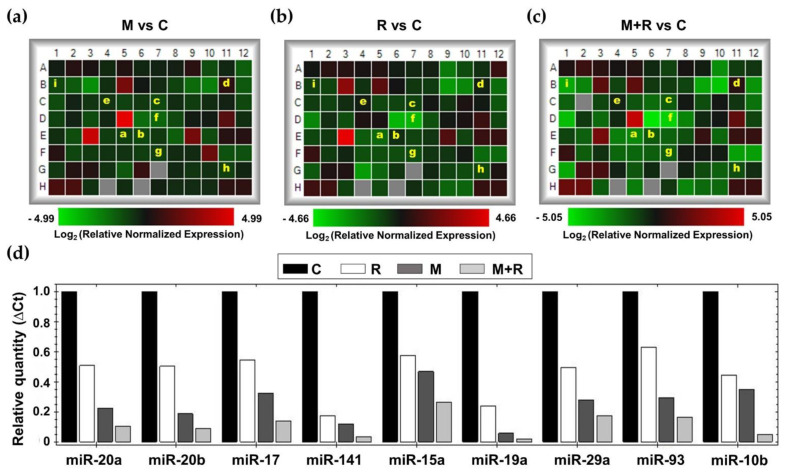
Effects of melatonin and radiation on breast cancer miRNA expression microarray. Total RNA from MCF-7 cells was extracted 4 h after radiation, reverse transcribed and used for RT-PCR analysis using Human Breast Cancer microarray (MIHS-109ZA). (**a**–**c**) Heatmaps of relative normalized expression between different treatments (**a**) M vs. C; (**b**) R vs. C; (**c**) M + R vs. C; (**d**) Bar chart of relative normalized expression (ΔCt) of selected miRNAs. a: miR20a; b: miR-20b; c: miR-17; d: miR-141; e: miR-15a; f: miR-19a; g: miR29a; h: miR-93; i: miR-10b. C: Control; M: Melatonin pre-treated cells (1 nM); R: Radiated cells (8 Gy); M + R: Melatonin pre-treated and radiated cells.

**Figure 5 biomedicines-10-01088-f005:**
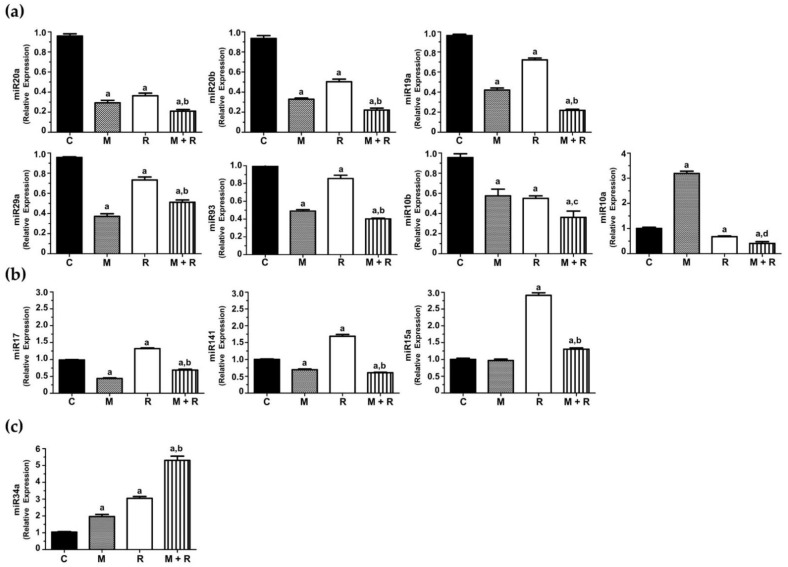
Changes in miRNA expression levels in MCF-7 cells. Cells were pre-treated with 1 nM melatonin for 1 week prior to being irradiated (8 Gy), and after 4 h total RNA was isolated and reverse transcribed; (**a**) RT-PCR analysis of selected miRNAs (miR-20a, miR-19a, miR-93, miR-20b, miR-29a, miR10a and miR10b) where the pre-treatment with melatonin potentiated the inhibitory effect of radiation; (**b**) RT-PCR analysis of selected miRNAs (miR-17, miR-141 and miR-15a) induced by radiation and counteracted by melatonin pre-treatment; (**c**) RT-PCR analysis of miR34a stimulated by radiation and melatonin. Data are expressed as relative normalized expression compared to control cells (mean ± S.E.M.). a, *p* ≤ 0.001 vs. C; b, *p*≤ 0.001 vs. R; c, *p* ≤ 0.05 vs. R; d, *p* ≤ 0.01 vs. R.

**Figure 6 biomedicines-10-01088-f006:**
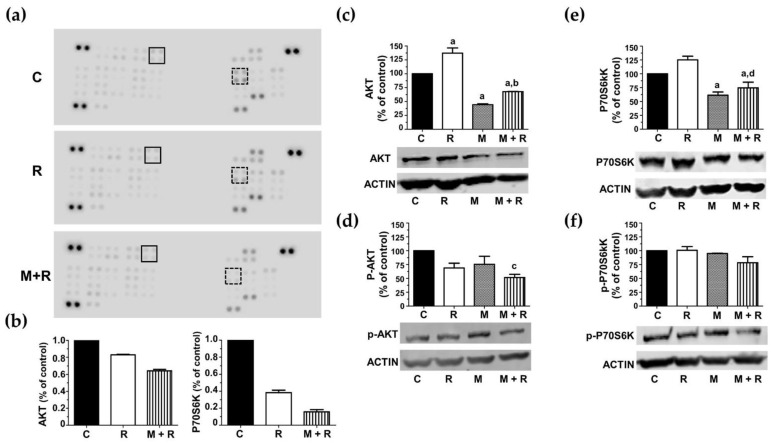
Effects of melatonin and radiation on the phosphorylation levels of Akt and p70S6 kinases. (**a**) Representative images of Proteome Profiler array blots assessed with protein extracts from MCF-7 control cells (C), radiated (R) or pre-treated with melatonin before radiation (M + R). Dots corresponding to Akt are shown in black square and p70S6K dots are shown in dotted square; (**b**) bar chart compiling quantification densitometry data of Akt dots and p70S6K dots, expressed as percentage of control cells; (**c**) western blot detection of total Akt; (**d**) western blot detection of phosphorylated Akt (Ser473); (**e**) western blot detection of total P70S6 kinase; (**f**) western blot detection of phosphorylated P70S6 kinase (Thr389). Bar charts from western blots are expressed as percentages of control non-radiated cells, using β-actin, as a loading control. a, *p* ≤ 0.01 vs. C; b, *p* ≤ 0.01 vs. R; c, *p* ≤ 0.05 vs. C; d, *p* ≤ 0.001 vs. R.

**Figure 7 biomedicines-10-01088-f007:**
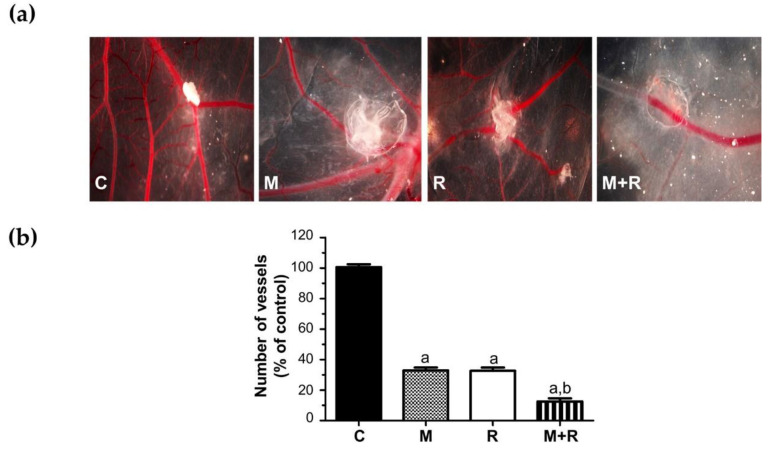
In vivo effect of melatonin pre-treatment on the radiation-induced actions on angiogenesis in the chorioallantoic membrane assay. MCF-7 cells pre-treated with melatonin and/or radiated were placed on the chorioallantoic membrane on day 11 of development. On day 14, newly formed blood vessels converging toward the alginates were counted at the microscopic level. (**a**) Representative images of CAM; (**b**) Vessel number data shown are expressed as mean ± S.E.M. a, *p* < 0.001 vs. C; b, *p* < 0.001 vs. R. C: Control; M: Melatonin pre-treated cells (1 nM); R: Radiated cells (8 Gy); M + R: Melatonin pre-treated and radiated cells.

**Table 1 biomedicines-10-01088-t001:** Primer sequences of genes tested by real-time RT-PCR.

Gene	Forward	Reverse
MUC1	ccaagagcactccattctcaatt	tggcatcagtcttggtgctatg
MYC	tgaggaggaacaagaagatg	atccagactctgaccttttg
BIRC5	tctccgcagtttcctcaaat	ggaccaccgcatctctacat
BCL2	cctttggaatggaagcttag	gagggaatgttttctccttg
ABCB1	gtacattaacatgatctggtc	cgttcatcagcttgatccgat
PTEN	aggtttcctctggtcctggt	cgacgggaagacaagttcat
NME1	agaagtctccacggatggt	agaaaggattccgccttgtt
RASSF1	atgaagtgcgtgaatgtatg	tgaggatcttgaaatctttat
CST6	ctcctctcagctcctaaag	tttattgtgacagatacggc
TP53	cctatgcttgtatggctaac	tagatccatgccttcttcttc
BAX2	aactggacagtaacatggag	ttgctggcaaagtagaaaag
BAD	atcatggaggcgctg	cttaaaggagtccacaaactc
CDKN1A	cagcatgacagatttctacc	cagggtatgtacatgaggag
PGR	gagagctcatcaaggcaattgg	caccatccctgccaatatcttg
RB1	accagatcatgtcagagag	taacctcccaatactccatc
SERPINE1	atccacagctgtcatagtc	cacttggcccatgaaaag
SNAI2	cagtgattatttccccgtatc	ccccaaagatgaggagtatc
SFN	tctgatcgtaggaattgagg	cacaggggaactttattgag
ERBB2	ccagcctgaatatgtgaac	ccccaaaggcaaaaacg
AKT1	aagtactctttccagaccc	ttctccagcttgaggtc
N3RC1	actgcttctctcttcagttc	gattttcaaccacttcatgc
PLAU	gctttaagattattgggggag	atgtagtcctccttctttgg
ATM	gagaaaagaagccgtgg	catcactgtcactgcac
APC	agaggtcatctcagaacaag	catgttgatttctcccactc
GATA3	cggtccagcacaggcagggagt	gagcccacaggcattgcagaca
β-ACTIN	tagcacagcctggatagcaa	aaatctggcaccacaccttc

**Table 2 biomedicines-10-01088-t002:** This table summarizes the distribution of breast cancer gene categories induced or repressed in MCF-7 cells treated with radiation (8 Gy) (R) or pre-treated with melatonin for one week (1 nM) before being irradiated (M + R). Pathway-focused gene expression profiling was performed using the Human Breast Cancer RT^2^ Profiler PCR Array. The number of up- and downregulated genes in each category in comparison to MCF-7 control cells (C) is indicated.

Breast Cancer Array	R vs. C		M + R vs. R	
	Up	Down	Up	Down
Angiogenesis, cell adhesion and proteases	7	14	8	20
Signal transduction	2	9	4	11
Apoptosis and Cell cycle	6	12	6	14
DNA damage and repair	5	3	2	4
Transcription factors	2	9	3	14

## Data Availability

The data that support the findings of this study are available from the corresponding author upon reasonable request.

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
