# Peer review of "Melatonin Modulation of Radiation-Induced Molecular Changes in MCF-7 Human Breast Cancer Cells"

_biomedicines, 2022, doi:10.3390/biomedicines10051088_

Round 1

Reviewer 1 Report

The authors of the present work studied the effect of melatonin and ionizing radiation on the MCF7 breast cancer cell line. The authors reported the molecular alterations induced by this molecule and radiation in terms of gene, miRNA and protein expression.   

The manuscript is well written but contains significant flaws and for this reason the article needs some major revisions.

The manuscript would benefit from the following:

  • In the study the authors used only one cell line. This aspect limits the translational value of the data obtained because they can be ascribed to MCF7 only. The possibilities to increase the relevance of the study are different. The authors should performed the same experiments on other cell lines or primary samples belonging to the same or different histotypes. Another options could be performed in silico analysis to confirm the results in bigger groups of samples.
  • The authors studied different molecular alteration induced by melatonin and ionizing radiation treatments. The technologies used are not really informative. Due to the low number of samples, I think that sequencing analysis could increase the information about the effect of these agents on MCF7 cells.
  • It is not really clear the aim of the angiogenesis experiments. It appears unnecessary and adds only a preliminary result. The authors should remove this paragraph or perform other experiments on the modulation of angiogenesis processes.
  • Some relevant work are missing. The following recent papers describing drug resistance mechanisms specific of MCF7 should be referenced: “Doxorubicin resistance in breast cancer cells is mediated by extracellular matrix proteins”. BMC Cancer. 2018 Jan 6;18(1):41. doi: 10.1186/s12885-017-3953-6. and “Lineage-specific mechanisms and drivers of breast cancer chemoresistance revealed by 3D biomimetic culture”. Mol Oncol. 2022 Feb;16(4):921-939. doi: 10.1002/1878-0261.13037.

Author Response

Editor-in-chief

Biomedicines

Manuscript ID: biomedicines-1694741

Type: Article

Title: Melatonin modulation of radiation-induced molecular changes in MCF-7 human breast cancer cells

Authors: carolina alonso-gonzález , cristina gonzález-abalde , javier menéndez-menéndez , alicia gonzález-gonzález , Alicia González-Cabeza * , virginia álvarez-garcía , samuel cos , Carlos Martínez-Campa *

Section: Cancer Biology and Therapeutics

Special Issue: New Insights in Radiotherapy

Guest Editor: Carlos Martínez-Campa

Thank you very much for the evaluation of our manuscript entitled: “Melatonin modulation of radiation-induced molecular changes in MCF-7 human breast cancer cells”. We are sending you the updated version of the revised manuscript, in the hope you can find it suitable now for publication in Biomedicines.

Please find below a point by point response to the reviewers´ comments:

Reviewer 1:

The authors of the present work studied the effect of melatonin and ionizing radiation on the MCF7 breast cancer cell line. The authors reported the molecular alterations induced by this molecule and radiation in terms of gene, miRNA and protein expression.   

The manuscript is well written but contains significant flaws and for this reason the article needs some major revisions.

The manuscript would benefit from the following:

  • In the study the authors used only one cell line. This aspect limits the translational value of the data obtained because they can be ascribed to MCF7 only. The possibilities to increase the relevance of the study are different. The authors should performed the same experiments on other cell lines or primary samples belonging to the same or different histotypes. Another options could be performed in silico analysis to confirm the results in bigger groups of samples.
  • The authors studied different molecular alteration induced by melatonin and ionizing radiation treatments. The technologies used are not really informative. Due to the low number of samples, I think that sequencing analysis could increase the information about the effect of these agents on MCF7 cells.

Response: We would like to thank the reviewer for its critical comments that hopefully will help to improve the quality of our manuscript. We really thank the reviewer for its valuable suggestions. Melatonin has been repeatedly reported to inhibit the growth of the estrogen-responsive human breast cancer cell-line MCF-7, but it does not have any significant effect in the growth of the MDA-MB-231 triple negative human breast cancer cell-line (Mao et al. J Pineal Res. 2014 Apr;56,3:246-53). Although the MT1-associated Ga proteins are expressed in MDA-MB-231 cells, it was reported that aberrant signaling downstream of the Gai proteins resulted in differential regulation of the ERK1/2 activity. In the other hand, whereas MCF-7 expresses both ER and wild-type p53, MDA-MB-231 lack of both proteins, and it has been described that only breast cancer cells with intact p53 can induce melatonin receptors (MTs) synthesis. Another study (Eck et al. Br J Cancer. 1998 Jun;77(12):2129-37.doi: 10.1038/bjc.1998.357) demonstrated an apoptotic effect of melatonin on MCF-7 cells but not in MDA-MB-231.

In our previous study: “Deciphering the molecular basis of melatonin protective effects on breast cancer cells treated with doxorubicin: TWIST1 a transcription factor involved in EMT and metastasis, a novel target of melatonin”, (Cancers (Basel). 2019 Jul 19;11(7):1011. doi: 10.3390/cancers11071011) we modified the first version of our article accordingly to a reviewers´s kind similar suggestion, and thus, we performed the experiments in boht the estrogen responsive MCF-7 and the triple-negative MDA-MB-231 cell-lines. Whereas melatonin showed efficacy enhancing the anti-proliferative effect of doxorubicin and inhibiting doxorubicin stimulation of MCF-7 cells migration and invasion, it did not show any of those actions in MDA-MB-231 cells. Nevertheless, we have tested the effect of a combination of radiation and melatonin in the triple-negative cell line, and, in our hands, radiation-mediated inhibition of proliferation was unaffected by melatonin´s treatment. Therfore, we focused our investigation on the effect of both treatments in the estrogen-responsive cell-line. Although we have previously studied the effect of radiation and melatonin in co-cultures of MCF-7 and fibroblasts (Int J Mol Sci. 2019 Aug 13;20(16):3935. doi: 10.3390/ijms20163935) we have also published other studies describing the effect of both treatments in proteins involved in double-strand DNA break repair only in MCF-7 cells.

  • It is not really clear the aim of the angiogenesis experiments. It appears unnecessary and adds only a preliminary result. The authors should remove this paragraph or perform other experiments on the modulation of angiogenesis processes.

We deeply hank the reviewer for its comment and unfortunately we do not have the possibility to debate with the reviewer about it. With all due respect to the opinion of the reviewer, we do believe that the angiogenesis experiments rendered very interesting results. In the gene expression profile and the miRNAs profile we have performed and analysed in this work, we have found several genes and microRNAs previously described as involved in EMT, angiogenesis and metastasis, which expression was altered by ionizing radiation and modulated by melatonin. Therefore, in our opinion, it is important to demonstrate that the inhibitory effect of radiation, melatonin and combination of both on angiogenesis takes account not only in the in vitro cultured cells but also in the CAM in vivo model. Anyway, if the reviewer still considers that the angiogenesis experiments do not clearly contribute to the main idea of the manuscript we are willing to cut this section of our work

  • Some relevant work are missing. The following recent papers describing drug resistance mechanisms specific of MCF7 should be referenced: “Doxorubicin resistance in breast cancer cells is mediated by extracellular matrix proteins”. BMC Cancer. 2018 Jan 6;18(1):41. doi: 10.1186/s12885-017-3953-6. and “Lineage-specific mechanisms and drivers of breast cancer chemoresistance revealed by 3D biomimetic culture”. Mol Oncol. 2022 Feb;16(4):921-939. doi: 10.1002/1878-0261.13037.

We thank the reviewer for its suggestion, of course we do agree that these articles are relevant and we agree to include them in our manuscript. The included references are highlighted in red in the references´ section in the manuscript

Reviewer 2 Report

The paper presents valuable and clinically relevant data. However, below proposed revisions may significantly improve visibility and overall quality  of the publication. 

  1. Keywords should be extended presenting items which will attract more interest of multi-professional groups such as "anti-cancer therapy sensitizer", "improved individual outcomes".
  2. Keeping in mind point 1, corresponding statements should be provided in Abstract and "Conclusions" with outlook on how presented data can be implemented in daily practice improving individual outcomes. For doing that below listed references might be supportive: A). Item: Natural substances as effective anti-cancer therapy sensitizers Flavonoids as an effective sensitizer for anti-cancer therapy: insights into multi-faceted mechanisms and applicability towards individualized patient profiles. 2021. doi: 10.1007/s13167-021-00242-5.                                                                B). patient stratification and phenotyping in overall breast cancer management; innovative concepts of predictive diagnostics and personalised treatment: Cell-free nucleic acid patterns in disease prediction and monitoring-hype or hope? 2020. doi: 10.1007/s13167-020-00226-x.  
  3. Introduction should be structured in a reader-friendly manner presenting sub-titles as messages from corresponding paragraphs such as "Current deficits in breast cancer management", "Natural substances as effective anti-cancer therapy sensitizers", "Multi-functional role of melatonin in anti-cancer protection" etc. Working hypothesis should be presented instead of paragraphs dedicated to the study achievements.     
  4. Discussion should be better structured (see point 3).
  5.  Conclusion is too should and should be extended by clinically relevant statements - see points 1 and 2.

Author Response

Editor-in-chief

Biomedicines

Manuscript ID: biomedicines-1694741

Type: Article

Title: Melatonin modulation of radiation-induced molecular changes in MCF-7 human breast cancer cells

Authors: carolina alonso-gonzález , cristina gonzález-abalde , javier menéndez-menéndez , alicia gonzález-gonzález , Alicia González-Cabeza * , virginia álvarez-garcía , samuel cos , Carlos Martínez-Campa *

Section: Cancer Biology and Therapeutics

Special Issue: New Insights in Radiotherapy

Guest Editor: Carlos Martínez-Campa

Thank you very much for the evaluation of our manuscript entitled: “Melatonin modulation of radiation-induced molecular changes in MCF-7 human breast cancer cells”. We are sending you the updated version of the revised manuscript, in the hope you can find it suitable now for publication in Biomedicines.

Please find below a point by point response to the reviewers´ comments:

Reviewer 2:

The paper presents valuable and clinically relevant data. However, below proposed revisions may significantly improve visibility and overall quality  of the publication. 

First than all, we would like to thank the reviewer for its positive assesment of our work and above all, for its critical comments that hopefully will help to improve the quality of our manuscript and, in the reviewer´s own words, to make the work more attractive not only to basic but also to clinical researchers.

  1. Keywords should be extended presenting items which will attract more interest of multi-professional groups such as "anti-cancer therapy sensitizer", "improved individual outcomes".

Considering the reviewer´s opinion, we have included new ítems as keywords to attract the interest of multpirpofesional groups. The new list of keywords now is: Melatonin, pineal gland, breast cancer, radiotherapy, radiosensitizer, radiotherapy adverse effects, anti-cancer agents, anti-cancer therapy sensitizer, anti-angiogenesis, improved individual outcomes, MCF-7 cells, gene expression, miRNA expression.

  1. Keeping in mind point 1, corresponding statements should be provided in Abstract and "Conclusions" with outlook on how presented data can be implemented in daily practice improving individual outcomes. For doing that below listed references might be supportive: A). Item: Natural substances as effective anti-cancer therapy sensitizers Flavonoids as an effective sensitizer for anti-cancer therapy: insights into multi-faceted mechanisms and applicability towards individualized patient profiles. 2021. doi: 10.1007/s13167-021-00242-5.      B). patient stratification and phenotyping in overall breast cancer management; innovative concepts of predictive diagnostics and personalised treatment: Cell-free nucleic acid patterns in disease prediction and monitoring-hype or hope? 2020. doi: 10.1007/s13167-020-00226-x.  

We really appreciate the reviewers´ recommendation and we completely agree with the idea that including these statements in the “abstract” and the “conclusions” section might help to attract the interest of both researchers and clinicians to our work. Therefore, paragraphs (highlighted in red color in the revised version) of the manuscript have been included attending the reviewers´suggestion in both sections.

  1. Introduction should be structured in a reader-friendly manner presenting sub-titles as messages from corresponding paragraphs such as "Current deficits in breast cancer management", "Natural substances as effective anti-cancer therapy sensitizers", "Multi-functional role of melatonin in anti-cancer protection" etc. Working hypothesis should be presented instead of paragraphs dedicated to the study achievements.   

We have taken into consideration the reviewers´opinion, and accordingly to the reviewers´s idea, we have included sub-titles as messages from corresponding paragraphs, and have highlighted a working hypothesis.

  1. Discussion should be better structured (see point 3).

Same as in 3, we have considered the reviewers´ kind suggestion and we have restructured the discusion as indicated.

  1.  Conclusion is too should and should be extended by clinically relevant statements - see points 1 and 2.

Accordingly to the reviewers´ recommendation, the “conclusion” section has been extended (highlighted in red in the new version of the manuscript) with a paragraph including the potential clinical benefits and the neccesity

Round 2

Reviewer 1 Report

The paper has been improved, most important issues rised are solved. The paper should be considered for pubblication in the present form.